# Prognostic Value of Blood-Based Inflammatory Markers in Cancer Patients Receiving Immune Checkpoint Inhibitors

**DOI:** 10.3390/cancers17010037

**Published:** 2024-12-26

**Authors:** Mustafa Murat Midik, Damla Gunenc, Pınar Fatma Acar, Burcak Saziye Karaca

**Affiliations:** Division of Medical Oncology, Department of Internal Medicine, Medical Faculty, Ege University, 35100 Izmir, Turkey; mustafa.murat.midik@ege.edu.tr (M.M.M.); damla.gunenc@ege.edu.tr (D.G.); fatma.pinar.acar@ege.edu.tr (P.F.A.)

**Keywords:** immune checkpoint inhibitors, inflammatory biomarkers, neutrophil-to-lymphocyte ratio (NLR), systemic immune-inflammation index (SII), cancer prognosis, overall survival (OS)

## Abstract

Immune checkpoint inhibitors (ICIs) play crucial roles in the treatment of various cancers and improve overall survival (OS) and progression-free survival (PFS). However, resistance is a major problem that limits the prognosis of ICI-based therapy. Determining inflammatory markers may ameliorate the benefits of ICI-based therapies. Peripheral blood cells and biochemical markers are big candidates as inflammatory markers. Therefore, this study aimed to evaluate and compare the prognostic value of blood-based inflammatory markers in ICI-based therapy.

## 1. Introduction

Immune checkpoint inhibitors (ICIs) have broadened treatment horizons across multiple cancer types, offering substantial survival benefits and prolonged response durations by targeting the programmed cell death protein 1 (PD-1), programmed death-ligand 1 (PD-L1), and cytotoxic T-lymphocyte-associated protein 4 (CTLA-4) to enhance anti-tumoral immune response [1]. Despite promising outcomes, objective response rates in tumors that typically respond better to immunotherapy, such as melanoma, lung, and renal cell cancer, can reach about 50% with combination therapies [2,3,4]. With single-agent treatments, these rates have remained around 20–40% [5,6,7]. This indicates that more than half of patients do not benefit from these agents, and some may lose this response during treatment due to the development of resistance to the therapy. Therefore, one of the most critical needs in the field is reliable and clinically accessible biomarkers that can predict primary or secondary resistance to ICIs in the quest to optimize and personalize cancer immunotherapy [8,9].

PD-L1 was the first biomarker to be used to predict the response to pembrolizumab [10,11]. However, its predictive value is limited, as PD-L1 expression can be induced by interferon and various immunological signaling pathways during treatment, making it less reliable than a standalone predictive biomarker. Other than PD-L1, the utility of various markers, including genetic biomarkers such as tumor mutation burden (TMB), IFN-γ gene expression profiles, mismatch repair system deficiency (dMMR), and high microsatellite instability (MSI-H), as well as surface markers including other inhibitory checkpoint receptors and physiological factors like gut microbiota, has been suggested for predicting response to ICIs [12,13,14,15]. However, the clinical application of these biomarkers is considerably restricted by several factors. The analysis of tissue samples is heavily influenced by tumor heterogeneity. The need for new biopsies for evaluating secondary resistance or longitudinal monitoring presents additional challenges. Furthermore, the high costs and limited accessibility of genomic methods, along with the lack of standardization for many of these biomarkers, further constrain their use in clinical practice.

Growing evidence suggests that the systemic inflammatory response is a critical determinant of tumor progression and patient survival across various malignancies. ICIs inherently affect various immune system components, making inflammatory markers potentially more valuable for predicting treatment outcomes. Among these, high levels of the neutrophil-to-lymphocyte ratio (NLR) and platelet-to-lymphocyte ratio (PLR) have been extensively documented to predict poor outcomes in patients with cancer [16,17,18]. Recent research has focused on broader evaluation methods for the inflammatory response that integrate more than two blood parameters, such as the systemic immune-inflammation index (SII), pan-immune inflammation value (PIV), systemic inflammation response index (SIRI), and Glasgow Prognostic Score (GPS) [19,20,21,22,23,24]. Furthermore, prognostic scoring systems that combine blood-based parameters, such as the Royal Marsden Hospital (RMH) score, with clinical parameters are increasingly being explored for their higher prognostic measurement power and potential clinical applicability [25].

However, the efficacy of these biomarkers in predicting ICI treatment outcomes remains under debate, with studies showing conflicting results and no consensus on optimal cut-off values. In this study, we aimed to further evaluate the prognostic significance of multiple inflammatory biomarkers among patients with a variety of types of cancer receiving ICI-based therapies.

## 2. Materials and Methods

### 2.1. Study Design and Patient Population

This retrospective, single-center study was conducted at the Ege University Medical Faculty Oncology Hospital outpatient clinic. We included adult patients who received ICIs between September 2012 and February 2023. The study was approved by the Ege University Ethics Committee (22-12.1T/10) and complied with the principles of the Declaration of Helsinki. Medical histories and laboratory data were retrieved from the patients’ electronic files. A total of 13,758 patients were screened, and 330 patients receiving any ICI treatments, such as anti-PD-1, anti-PD-L1, or anti-CTLA-4, were identified. Patients who were under 18 years of age, received less than three cycles of treatment, were diagnosed with secondary malignancies, or were pregnant were excluded. A total of 226 of the 330 patients who met the inclusion/exclusion criteria were enrolled in the study.

### 2.2. Data Collection and Definition of Inflammatory Biomarkers

Demographic and clinicopathologic characteristics, tumor–node–metastasis (TNM) stage, ICI-related adverse events (IrAE), previous treatment history, BRAF status for patients with melanoma, PD-L1 status for patients with non-small cell lung cancer (NSCLC), date of ICI treatment initiation, date of progression, and death or last follow-up were recorded. To ensure the quality and reliability of the data, two investigators (D.G. and F.P.A.) conducted a chart review of randomly selected cases (10% of the total cohort) to verify the accuracy of the demographic, clinicopathologic, and laboratory values. Furthermore, the dataset was thoroughly reviewed for inconsistencies and outliers, and any discrepancies were resolved through consensus. Baseline laboratory values recorded up to seven days prior to the initiation of ICI therapy, including albumin, C-reactive protein (CRP), lactate dehydrogenase (LDH) levels, and counts of lymphocytes (L), neutrophils (N), hemoglobin (Hb), monocytes (M), eosinophils (E), and platelets (P). The NLR, eosinophil/lymphocyte ratio (ELR), PIV, SII, and SIRI were calculated, respectively, as NLR = N/L, ELR = E/L, PIV = (N × P × M)/L, SII = P × N/L, and SIRI = M × N/L.

### 2.3. Statistical Analysis

IBM SPSS version 27 and Jamovi statistical software version 2.3.28 were utilized for data analysis. Parametric tests were used for non-normality testing in accordance with the Central Limit Theorem. Continuous variables were presented as mean, standard deviation, median, minimum, and maximum values. Categorical variables were characterized by frequency and percentage values. The relationship between categorical variables was assessed using the Chi-square test. For diagnostic studies, the receiver operating characteristic (ROC) curve is frequently used for continuous variables, as it evaluates a binary outcome. By contrast, prognostic studies typically assess time-to-event outcomes, such as survival rates, introducing the follow-up time as an additional variable. Thus, we preferred to use maximally selected rank statistics, which incorporate follow-up time for more accurate calculation of cut-offs for NLR, ELR, PIV, SII, SIRI, LDH, CRP, and albumin. This method divides patients into two groups on the basis of the most significant statistical differences between them to identify the maximum of the standardized statistics for all possible cut points [26].

The primary clinical outcome was overall survival (OS). OS was defined as the time from the date of ICI treatment initiation to the date of death from any cause or last follow-up. Progression-free survival (PFS) was measured from the ICI treatment initiation date to disease progression, death, or last follow-up. Response to treatment was determined using the Response Evaluation Criteria in Solid Tumors, version 1.1. OS and PFS curves were estimated via the Kaplan–Meier method using the log-rank test. The median follow-up time for PFS and OS was calculated using a reverse Kaplan–Meier estimate. Potential factors affecting OS and PFS were assessed in univariate Cox regression analysis, and the hazard ratio was provided with a 95% confidence interval (CI). The ones with *p* < 0.05 were included in the multivariate analysis. Spearman’s correlation tests were used to assess the pairwise relationship among inflammatory markers. The predictive accuracy of the Cox regression models was assessed by Harrel’s concordance index (c-index), which is the probability of concordance between predicted and observed survival, ranging from 0.5 (no discrimination) to 1 (perfect discrimination).

## 3. Results

### 3.1. Summary of Patient Characteristics

We enrolled 226 cancer patients receiving ICI-based therapies. The median age at the time of the ICI-based therapy was 56.5 (IQR: 46.8–65) years. Overall, 63.3% (n = 143) were men, and 36.7% (n = 83) were women. The most common histologies were melanoma (46%), NSCLC (20.4%), RCC (15.9%), and breast (5.8%). The remaining 27 patients (11.9%) had various cancers, including head and neck (2.2%), bladder (1.8%), colorectal (1.8%), endometrium (1.3%), soft tissue (0.9%), gastric (0.9%), SCLC (0.9%), ovarian (0.9%), HCC (0.9%), and thymoma (0.4%). Most patients (97.4%) included in the study had stage IV disease. The majority of patients (65.9%) had received at least one prior line of systemic treatment, with the remaining (34.1%) having no prior exposure to any systemic treatment. Half of the study population (n = 113) received single-agent nivolumab. Including those who were treated with ipilimumab (n = 35), pembrolizumab (n = 19), avelumab (n = 14), atezolizumab (n = 7), and spartalizumab (n = 2), a total of 190 (84.1%) patients received ICI monotherapy. In addition, 17 patients (8%) received dual ICI therapy, while 19 (8.4%) received a combination of ICI with either chemotherapy or a TKI. The demographic information, disease, and treatment characteristics of the study population are presented in Table 1.

### 3.2. Optimal Cut-Off Values for NLR, PIV, SII, SIRI, LDH, CRP, and Albumin

The baseline blood counts and blood-based inflammatory markers of the study population are summarized in Appendix A. Blood-based biomarkers and systemic inflammatory indexes were dichotomized according to the optimal cut-off based on maximally selected rank statistics using OS as the outcome variable (Figure 1). The optimal cut-off values for NLR, PIV, SII, SIRI, LDH, CRP, and albumin were calculated as 2.62, 1312, 969, 1.93, 334 U/L, 12 mg/dL, and 39.2 g/L, respectively (*p* < 0.001 for each). ELR was excluded from the further analysis, as a statistically significant cut-off value could not be determined (*p* = 0.846).

### 3.3. Association of Blood-Based Inflammatory Markers and Clinical Outcomes

The median follow-up for survivors was 55.6 months. The median PFS in all 226 patients was 9.2 months (95% CI, 8.1–10.2 months), and the median OS was 20.8 months (95% CI, 17.3–24.4 months). For 2, 3, and 5 years, PFS and OS were 29.5%, 27.4%, and 23.1% and 43.5%, 33.2%, and 26.4%, respectively. OS was stratified by dichotomous inflammatory markers. The median OS was shorter for patients who had high NLR, PIV, SII, SIRI, LDH, and CRP values and low albumin levels at baseline (log-rank *p* ≤ 0.001). Figure 2 provides Kaplan–Meier curves for the overall population and stratified OS regarding inflammatory markers.

Univariate analyses were carried out to identify variables associated with PFS and OS. Female sex and earlier stages were found to be associated with better PFS and OS (HR = 0.7, 95% CI 0.5–0.98; HR = 0.66, 95% CI 0.46–0.94, respectively). Patients who had a history of any treatment had worse OS (HR = 1.53, 95% CI 1.07–2.19). Regarding the inflammatory variables, elevated CRP and LDH, high NLR, high PIV, high SII, high SIRI, and low albumin levels were associated with worse PFS and OS (Table 2).

There was an expected collinearity between the NLR, PIV, SII, and SIRI due to mathematical coupling in the calculations. For example, both the NLR and SII rely on the neutrophil and lymphocyte ratio, which means they are influenced in the same direction by changes in these parameters. We further evaluated the correlation between all inflammatory markers, including LDH, albumin, and CRP (Appendix A, Appendix A). As we observed a strong correlation between these values, we established a base model (Model-1) consisting of significant risk factors determined in the univariate analysis, including sex, stage, tumor type, and previous treatment history. Then, inflammatory markers were added to Model-1 separately to avoid multicollinearity. Age, treatment line, and irAE history were excluded from the multivariable model, as they were not significant in the univariable analyses.

The results, shown in Table 3, revealed that high CRP (HR: 2.84, 95% CI: 1.81–4.53, *p* < 0.001), high LDH (HR: 3.79, 95% CI: 2.43–5.79, *p* < 0.001), high NLR (HR: 2.12, 95% CI: 1.48–3.06, *p* < 0.001), high PIV (HR: 2.51, 95% CI: 1.63–3.77, *p* < 0.001), high SII (HR: 2.05, 95% CI: 1.44–2.93, *p* < 0.001), and high SIRI (HR: 1.76, 95% CI: 1.23–2.52, *p* = 0.002) were independently associated with worse OS. By contrast, high albumin was associated with improved OS (HR: 0.46, 95% CI: 0.33–0.65, *p* < 0.001).

The addition of each inflammatory marker improved model performance with a higher c-index compared to Model-1. The multivariable c-index was highest for the model with CRP (0.762), followed by NLR (0.694), LDH, and SIRI (0.684, for each).

## 4. Discussion

Resistance to ICI treatment is frequently observed, prompting numerous studies focusing on potential predictive and prognostic markers for appropriate patient–treatment matching and improved outcomes. Thus, in the current study, we evaluated the prognostic value of blood-based inflammatory markers in patients with cancer receiving ICIs, as these markers are easily accessible and hold the potential to reflect treatment outcomes.

Among these markers, historically defined acute phase reactants such as LDH, CRP, and albumin are well-known indicators of systemic inflammation. Their prognostic value was shown for a variety of diseases, including patients with cancer [27,28,29,30,31]. In our study, high CRP and LDH levels were strongly associated with worse OS. Conversely, as albumin is a negative acute phase protein and an indicator of nutritional status, patients with low albumin levels had shorter OS rates. The findings of our study are in agreement with the literature. However, individual markers such as CRP, LDH, and albumin are influenced by a variety of non-tumor-related factors, resulting in low specificity. To overcome this limitation, composite indexes like NLR, PIV, SII, and SIRI have been suggested.

NLR is one of the most frequently documented blood-based prognostic inflammatory indices in patients with cancer. It is thought to reflect the balance between the inflammatory environment caused by the tumor and the host immune response. There is a convincing number of studies showing its prognostic significance in patients with cancer receiving chemotherapy and ICIs [17,18,32,33]. Although the prognostic impact of NLR has been demonstrated across a broad spectrum of cancer types, in an umbrella study including 204 meta-analyses, the most substantial evidence was observed in prostate, non-muscle-invasive bladder, and nasopharyngeal cancers, among others [16].

In the search for more accurate reflections of the complex immune network, new inflammatory indexes incorporating several markers have been proposed. In this context, SII, which combines NLR and the platelet–lymphocyte ratio, was first introduced by Hu et al. in 2014 [34]. This trend continued with the introduction of other prognostic indexes, such as SIRI and PIV, which were also suggested for prognostic evaluation [35,36]. Several studies have demonstrated that higher values of PIV, SII, and SIRI are associated with poor OS in patients with cancer treated with ICIs [20,23,37,38,39,40]. Our findings are in line with the growing body of literature showing that low values of NLR, SII, SIRI, and PIV are correlated with better OS.

While many studies support the prognostic utility of these inflammatory markers, there are also conflicting data. For example, in a study evaluating PIV and SII in patients with advanced melanoma receiving ICIs, there was no significant correlation with clinical outcomes. Additionally, although the prognostic impact of NLR has been demonstrated across a broad spectrum of cancer types, in an umbrella study including 204 meta-analyses, the most substantial evidence was observed in the prostate, non-muscle-invasive bladder, and nasopharyngeal cancers, among others [16]. Furthermore, there is substantial variability in the literature regarding the cut-off values of these markers. For NLR, typical cut-off values range from 2 to 5, and for SIRI, they range between 0.5 and 3.5 [23,32]. The values for SII reported in the literature show a much wider spectrum, with values such as 197 and 1375 being reported [20].

Variations in study populations, cancer types, and treatment regimens are considered possible reasons for these differing results. However, there are also two significant methodological issues specific to these studies investigating the prognostic significance of inflammatory markers. First, there is inherent multicollinearity among the newly proposed inflammatory markers, such as PIV, SIRI, and SII, as they share common components in their calculations. This makes it challenging to compare these markers with each other within the same study population. In several studies evaluating multiple blood-based markers, prognostic variables in univariate analyses did not retain their prognostic significance in the multivariate model [40,41]. Recognizing this concern, we evaluated these markers separately in the multivariate analysis. Another key limitation is using the standard ROC method while determining cut-offs, leading to misinterpretation of time-dependent outcomes such as survival [42,43].

Our study has several limitations that warrant consideration. First, this is a retrospective analysis, which inherently carries the risk of selection bias and missing data. Second, the study was conducted at a single center, which may limit the generalizability of our findings. Third, we did not explore the temporal changes of these inflammatory markers during the course of ICI-based therapies, as changes during treatment may offer additional prognostic insights. Additionally, although the study population consists of a heterogeneous group of patients with different cancer types and receiving various ICI combinations, which could initially be considered a limitation, this extensive evaluation highlights the broad applicability of these markers as well. Although approximately 95% of the patients included in our study had metastatic cancer and tumor stage was added as a confounder in the analysis, the inclusion of patients with different stages remains a limitation in terms of heterogeneity.

Despite some limitations, our work contributes to the literature by including a large diverse cohort of patients, demonstrating the prognostic value of multiple inflammatory markers, and highlighting methodological issues that should be addressed in future research.

## 5. Conclusions

Inflammatory markers are easily accessible, cost-effective tools with significant prognostic potential in patients with cancer treated with ICIs. Our results suggest that routine monitoring of CRP, NLR, SII, and related indices could aid in treatment stratification. Further studies are needed to validate these findings in larger, multi-center cohorts and investigate the interplay between temporal changes in these markers and ICI response.

## Figures and Tables

**Figure 1 cancers-17-00037-f001:**
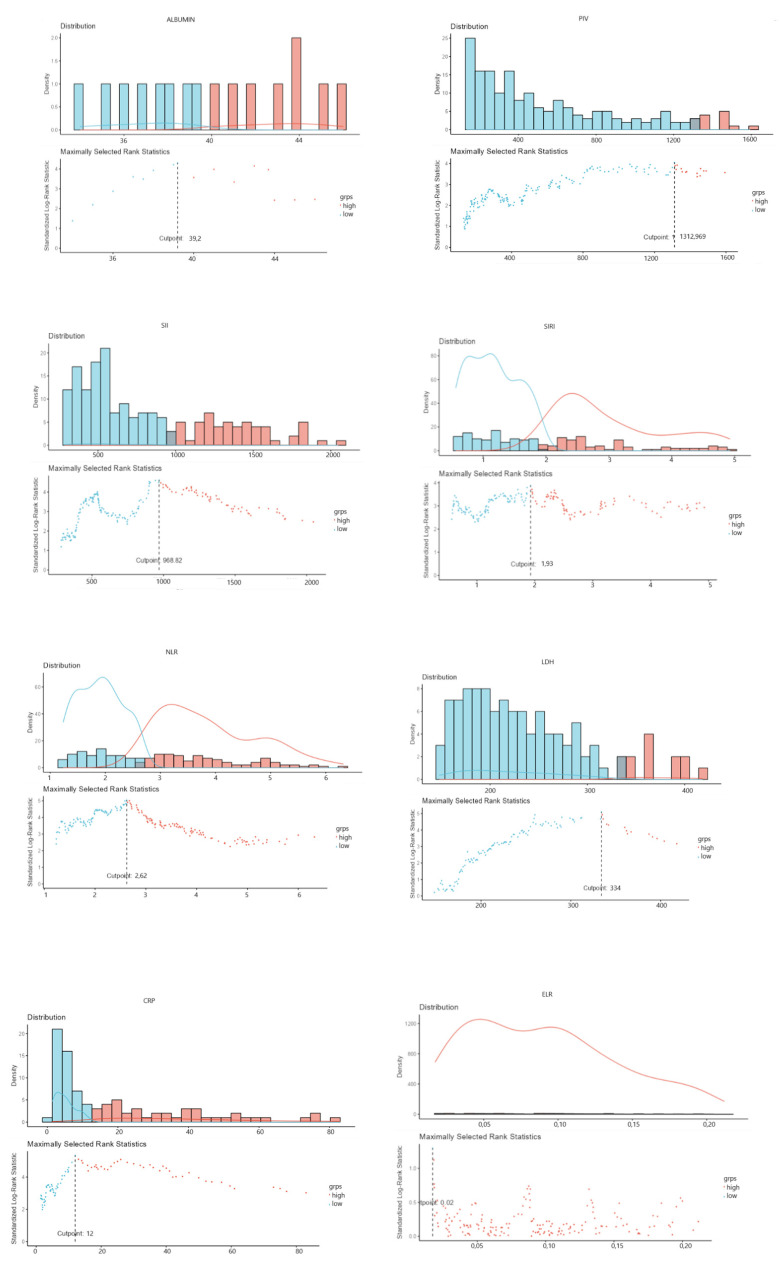
Scatter plots of the maximally selected rank statistic evaluating optimal cut-off values for NLR, PIV, SII, SIRI, LDH, CRP, albumin, and ELR.

**Figure 2 cancers-17-00037-f002:**
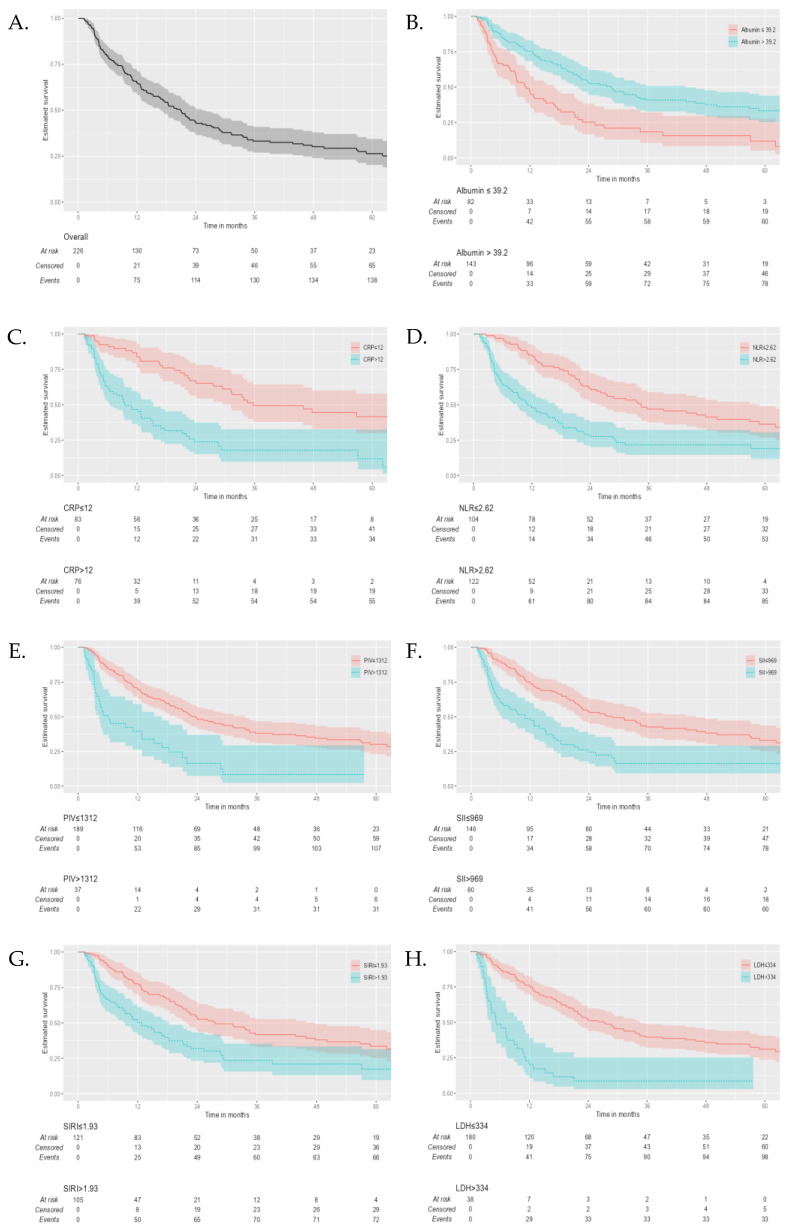
Kaplan–Meier curves of overall survival (OS) for (**A**) the overall study population and high versus low baseline (**B**) albumin values, (**C**) CRP, (**D**) neutrophil-to-lymphocyte ratios (NLRs), (**E**) pan-immune inflammation values (PIVs), (**F**) systemic immune-inflammation indexes (SIIs), (**G**) systemic inflammation response indexes (SIRIs), and (**H**) LDH values.

**Table 1 cancers-17-00037-t001:** Patient characteristics (n = 226).

		$\bar{x}$ ± SD	Median (Min–Max)
**Age**		55.5 ± 13	56.5 (18–83)
		**N**	**(%)**
**Sex**	Men	143	63.3
	Women	83	36.7
**Histology**	Melanoma	104	46
	NSCLC	46	20.4
	RCC	36	15.9
	Breast	13	5.8
	Head-Neck	5	2.2
	Bladder	4	1.8
	Colorectal	4	1.8
	Endometrium	3	1.3
	Other (Soft Tissue, Gastric, SCLC, Ovarian, HCC, Thymoma)	11	4.9
**Stage**	II	3	1.3
	III	9	4.0
	IV	214	94.7
**BRAF (melanoma)**	Negative	70	67.3
	Positive	34	32.7
**PD-L1 (NSCLC)**	Negative	6	13
	Positive	13	28.3
	Unknown	27	58.7
**Treatment regimen**	** ICI monotherapy **	190	84.1
	Nivolumab	113	50
	Ipilimumab	35	15.5
	Pembrolizumab	19	8.4
	Avelumab	14	6.2
	Atezolizumab	7	3.1
	Spartalizumab	2	0.9
	** Dual ICI **	17	7.5
	Ipilimumab + Nivolumab	15	6.6
	Pembrolizumab + Ipilimumab	2	0.9
	** ICI + ChT **	18	8
	Pembrolizumab + ChT	13	5.8
	Atezolizumab + ChT	5	2.2
	** ICI + TKI **		
	Pembrolizumab + Lenvatinib	1	0.4
**History of Previous Treatment**	No	77	34.1
	Yes	149	65.9
**Number of prior systemic therapies**	0	77	34.1
	1	89	39.4
	2	38	16.8
	3≥	22	9.7
**IrAE**	No	182	80.5
	Yes	44	19.5

ChT, Chemotherapy; HCC, hepatocellular cancer; ICI, immune checkpoint inhibitor; IrAE, ICI-related adverse events; NSCLC, non-small cell lung cancer; PD-L1, programmed cell death protein ligand-1; SCLC, small cell lung cancer.

**Table 2 cancers-17-00037-t002:** Univariable Cox proportional hazards regression PFS and OS.

Variables		N	Univariate			
			PFS		OS	
			HR (95% CI)	*p*-Value	HR (95% CI)	*p*-Value
**Age**	≤60	151	Reference		Reference	
	>60	75	1.125 (0.807–1.567)	0.487	1.304 (0.925–1.837)	0.130
**Sex**	Men	143	Reference		Reference	
	Women	83	0.697 (0.499–0.975)	**0.035**	0.659 (0.463–0.938)	**0.020**
**Histology**	Melanoma	104	Reference		Reference	
	NSCLC	46	1.484 (0.916–2.405)	0.109	1.194 (0.726–1.966)	0.485
	RCC	36	1.991 (1.149–3.449)	**0.014**	2.017 (1.162–3.500)	**0.013**
	Other	40	1.005 (0.556–1.816)	0.988	0.696 (0.364–1.330)	0.273
**Stage**	IV	214	Reference		Reference	
	II/III	12	0.337 (0.125–0.910)	**0.032**	0.280 (0.089–0.880)	**0.029**
**Previous Treatment**	No	77	Reference		Reference	
	Yes	149	1.329 (0.947–1.866)	0.100	1.534 (1.073–2.194)	**0.019**
**Treatment Line**	≤1	166	Reference		Reference	
	>1	60	858 (0.593–1.242)	0.418	1.067 (0.730–1.558)	0.738
**IrAE**	No	44	Reference		Reference	
	Yes	182	0.801 (0.542–1.183)	0.265	0.659 (0.427–1.015)	0.059
**CRP (mg/L)**	Low (≤12)	83	Reference		Reference	
	High (>12)	76	2.876 (1.910–4.330)	**<0.001**	3.341 (2.162–5.163)	**<0.001**
**Albumin (g/L)**	Low (≤39.2)	82	Reference		Reference	
	High (>39.2)	143	0.601 (0.434–0.831)	**0.002**	0.445 (0.317–0.625)	**<0.001**
**LDH (U/L)**	Low (≤334)	180	Reference		Reference	
	High (>334)	38	2.904 (1.947–4.332)	**<0.001**	4.045 (2.693–6.077)	**<0.001**
**NLR**	Low (≤2.62)	104	Reference		Reference	
	High (>2.62)	122	1.902 (1.377–2.629)	**<0.001**	2.429 (1.722–3.425)	**<0.001**
**PIV**	Low (≤1312)	189	Reference		Reference	
	High (>1312)	37	1.920 (1.285–2.869)	**0.001**	2.792 (1.860–4.193)	**<0.001**
**SII**	Low (≤969)	146	Reference		Reference	
	High (>969)	80	1.815 (1.312–2.511)	**<0.001**	2.397 (1.706–3.369)	**<0.001**
**SIRI**	Low (≤1.93)	121	Reference		Reference	
	High (>1.93)	105	1.551 (1.129–2.131)	**0.007**	1.981 (1.418–2.767)	**<0.001**

IrAE, ICI-related adverse events; LDH, lactate dehydrogenase; NLR, neutrophil/lymphocyte ratio; NSCLC, non-small cell lung cancer; PIV, pan-immune-inflammation value; RCC, renal cell carcinoma; SII, systematic immune-inflammation index; SIRI, systematic inflammatory response index.

**Table 3 cancers-17-00037-t003:** Multivariate Cox proportional hazards regression of inflammatory biomarkers.

Variables		OS		Model Performance *
		HR (95% CI)	*p* Value	
**Sex**	Men	Reference		**Model-1** (Variables: Sex, Histology, Stage, and Previous Treatment) (n = 226) c-index = 0.628; LR = 26.00
	Women	0.721 (0.491–1.047)	0.0898	
**Histology**	Melanoma	Reference		
	NSCLC	1.337 (0.855–2.063)	0.1948	
	RCC	0.497 (0.281–0.835)	**0.0115**	
	Other	0.991 (0.581–1.617)	0.9718	
**Stage**	IV	Reference		
	II/III	0.363 (0.087–1.015)	0.0939	
**Previous Treatment**	No	Reference		
	Yes	1.299 (0.892–1.922)	0.1805	
**CRP (mg/L)**	Low (≤12)	Reference		**Model-1 + CRP** (n = 159) c-index = 0.724; LR = 21.08
	High (>12)	2.841 (1.811–4.529)	**<0.001**	
**Albumin (g/L)**	Low (≤39.2)	Reference		**Model-1 + Albumin** (n = 225) c-index = 0.672; LR = 18.22
	High (>39.2)	0.461 (0.326–0.654)	**<0.001**	
**LDH (U/L)**	Low (≤334)	Reference		**Model-1 + LDH** (n = 218) c-index = 0.684; LR = 30.60
	High (>334)	3.788 (2.430–5.789)	**<0.001**	
**NLR**	Low (≤2.62)	Reference		**Model-1 + NLR** (n = 226) c-index = 0.694; LR = 17.11
	High (>2.62)	2.124 (1.484–3.062)	**<0.001**	
**PIV**	Low (≤1312)	Reference		**Model-1 + PIV** (n = 226) c-index = 0.669; LR = 15.88
	High (>1312)	2.508 (1.626–3.770)	**<0.001**	
**SII**	Low (≤969)	Reference		**Model-1 + SII** (n = 226) c-index = 0.684; LR = 15.23
	High (>969)	2.054 (1.436–2.929)	**<0.001**	
**SIRI**	Low (≤1.93)	Reference		**Model-1 + SIRI** (n = 226) c-index = 0.672; LR = 9.380
	High (>1.93)	1.755 (1.225–2.518)	**0.002**	

LDH, lactate dehydrogenase; LR, log-likelihood ratio; NLR, neutrophil/lymphocyte ratio; PIV, pan-immune-inflammation value; SII, systematic immune-inflammation index; SIRI, systematic inflammatory response index. * Prediction compared with Model-1 (only Model-1 compared with null model).

## Data Availability

The data that support the findings of this study are available from the corresponding author upon request.

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
