# Peer review of "Prognostic Value of Blood-Based Inflammatory Markers in Cancer Patients Receiving Immune Checkpoint Inhibitors"

_cancers, 2024, doi:10.3390/cancers17010037_

Round 1

Reviewer 1 Report

Comments and Suggestions for Authors

Dear authors, I found your article very interesting. As far as I know it is one of the first that tries to find predictive response biomarkers to ICI treatments (efficient anti-cancer medicine, albeit very expensive), With all the method's limitations that you had excellently stated, I still find your work very valuable because it points out several biological parameters (both accessible and cheap) that a clinician can easily follow in order to stratify and/or follow the treatment. All in all, congratulations!

Author Response

Dear Reviewer,

Thank you very much for your positive and encouraging feedback regarding our manuscript. We are delighted to hear that you found our study both interesting and valuable, particularly in its effort to identify predictive response biomarkers for ICI treatments.

Your kind words motivate us to further explore and refine these findings in future studies, ensuring they can have a tangible impact on clinical practice. Once again, we thank you for your thoughtful and insightful review.

Kind regards

Reviewer 2 Report

Comments and Suggestions for Authors

The study evaluates the prognostic significance of various inflammatory markers, including neutrophil-to-lymphocyte ratio (NLR), systemic immune-inflammation index (SII), pan-immune inflammation value (PIV), and systemic inflammation response index (SIRI), in cancer patients undergoing immune checkpoint inhibitor (ICI)-based therapies. The text is well written and easy to read and follow it. I would like to offer the following points for consideration by the authors towards the improvement of the manuscript:

1-  Please incorporate a comparative discussion of the other inflammatory markers like Royal Marsden Hospital (RMH) score and the Glasgow Prognostic Score (GPS).

2- Please move the Kaplan-Meier plots currently in the supplementary materials into the main manuscript to enhance the clarity and impact of the results.

3-  The manuscript includes a heterogeneous patient population comprising various cancer types (e.g., melanoma, NSCLC, RCC) and treatment regimens (monotherapy vs. combination therapy). Please include a detailed subgroup analysis based on cancer types and treatment regimens. This would provide insights into whether the prognostic value of inflammatory markers varies across specific cancers or treatment approaches.

4- Please clarify certain terms, such as "expected collinearity," to make the results accessible to a broader audience

Author Response

Dear Reviewer 2,

We are grateful for your valuable and constructive feedback, which has provided us with the opportunity to improve our manuscript.

Comment 1-  Please incorporate a comparative discussion of the other inflammatory markers like Royal Marsden Hospital (RMH) score and the Glasgow Prognostic Score (GPS).

Response :1 :Comparative Discussion of RMH and GPS Scores:
We appreciate your suggestion to incorporate a discussion on the RMH and Glasgow Prognostic Score. Given the relevance and importance of these scoring systems in prognostic evaluations, we have added a section to our study. However, literature is lacking on comparison of these indexes, as we stated and highlighted for future aspects, and we do not have any data regarding these two indexes to make a comment on a comparative discussion.

Comment 2- Please move the Kaplan-Meier plots currently in the supplementary materials into the main manuscript to enhance the clarity and impact of the results.

Responce 2: Kaplan-Meier Plots in the Main Manuscript:
We agree that including Kaplan-Meier plots in the main text enhances the clarity and impact of the results by better illustrating the prognostic significance of the markers. We have incorporated the Kaplan-Meier plots from the supplementary materials into the main manuscript, as suggested.

Comment 3-  The manuscript includes a heterogeneous patient population comprising various cancer types (e.g., melanoma, NSCLC, RCC) and treatment regimens (monotherapy vs. combination therapy). Please include a detailed subgroup analysis based on cancer types and treatment regimens. This would provide insights into whether the prognostic value of inflammatory markers varies across specific cancers or treatment approaches.

Responce 3- Subgroup Analysis by Cancer Types and Treatment Regimens:
The heterogeneity of the patient population, and treatment regimens is indeed a limitation of our study, as noted in the manuscript. However, we believe that the observation of prognostic significance of these markers across a broad, heterogeneous population underscores their potential applicability regardless of cancer type.

Based on your suggestion, we performed subgroup analyses for four subgroups as; 1-melanoma, 2-RCC, 3-NSCLC, and 4- other cancers with univariate Cox regression. While markers like NLR, PIV, SII, and SIRI retained significance in melanoma, they were not significant in NSCLC (NLR, PIV, SII, SIRI) or RCC (NLR, SIRI). In the group of other cancers, only CRP remained significant. However, since the cut-off values were determined based on the overall population, we have doubts about the validity and applicability of these results to specific subgroups. For instance, none of the RCC patients had high LDH values according to the study cut-off, and in other cancers, only nine patients fell into the high LDH group. This disparity limits the reliability of applying these cut-offs and smaller sample sizes raise another challenge and limitation to our analysis. Additionally, subgroup analyses for treatment regimens (e.g., ICI with chemotherapy) were not feasible as patients receiving combination therapy represented only 8% of the population, making statistical interpretation unreliable.

Comment 4- Please clarify certain terms, such as "expected collinearity," to make the results accessible to a broader audience

Responce 4- Clarification of "Expected Collinearity":
We recognize that "expected collinearity" is a subjective term that might cause confusion for a broader audience. Based on your suggestion, we have clarified this terminology and provided a detailed explanation to enhance accessibility and understanding.

Once again, we thank you for your insightful suggestions, which have significantly improved the quality of our manuscript. We hope that our responses and the revisions made will meet your expectations.

Reviewer 3 Report

Comments and Suggestions for Authors

In this study, the authors evaluated the prognostic value of multiple inflammatory markers in cancer patients receiving immune checkpoint inhibitors (ICIs)-based therapies. By using the Kaplan-Meier method and Cox regression analysis, they found High NLR, PIV, SII, SIRI, LDH, and CRP, as well as low albumin levels, were associated with worse overall survival (OS) and progression-free survival (PFS). In multivariate analysis, they found high CRP, LDH, NLR, PIV, and SII independently predicted worse OS. Their findings confirmed the prognostic utility of several inflammatory biomarkers in cancer patients receiving ICIs, highlighting their potential for treatment stratification.

This study is well-designed for including different cancer patients and selecting inflammatory markers which are easily accessible, cost-effective tools. The manuscript is presented well and clearly. Below are my suggestions:

#1. Except c-index, please perform (ROC) curve and provide AUC value to verify the model.

#2.  Please try the combination of Model-1 and all inflammatory markers (CRP, Albumin, LDH, NLR, PIV, SII, SIRI) to get higher c-index.

#3. The authors could try to provide nomograms to better demonstrate the predictive function.

Author Response

Dear Reviewer-3,

We would like to express our gratitude for your thoughtful and constructive feedback. We are pleased that you found our study well-designed, and we greatly appreciate the opportunity to discuss the statistical and modeling challenges we encountered while conducting this research. We have carefully considered your suggestions, and we would like to address them as follows:

Comment 1. Except c-index, please perform (ROC) curve and provide AUC value to verify the model.

Response 1: Regarding the ROC curve and AUC value suggestion:
We selected OS as the endpoint in our study due to the inherent challenges of retrospective data reliability and the limitations of retrospective evaluation of RECIST criteria especially when applied to immunotherapy compared to cytotoxic treatments. While ROC curves are a valuable tool for dichotomous outcome analysis such as best response rates, they do not fully capture time-to-event outcomes such as OS. For this reason, we avoided using ROC curves for both cut-off determination and prognostic impact of the markers.

Comment 2.  Please try the combination of Model-1 and all inflammatory markers (CRP, Albumin, LDH, NLR, PIV, SII, SIRI) to get higher c-index.

Response 2: Combining Model-1 and all inflammatory markers:
As you rightly suggested, combining multiple inflammatory markers might theoretically improve the c-index. However, we faced significant multicollinearity issues during our analysis when we tried to combine all of them. For example, NLR (Neutrophil / Lymphocyte) and SII (Platelet × Neutrophil / Lymphocyte) both rely on neutrophil, and lymphocyte ratio, which means they are influenced in the same direction by changes in these parameters. This leads to mutual masking effects in multivariate analyses. To support this, we included supplementary results demonstrating high correlation among these biomarkers. These findings explain why we could not combine all available markers into a single model without risking overfitting.

Comment 3. The authors could try to provide nomograms to better demonstrate the predictive function.

Response 3: Providing nomograms:
Initially, we did explore the use of nomograms. However, our primary focus was to assess the individual prognostic power of inflammatory biomarkers, rather than creating a composite prognostic model or scoring system. Furthermore, as we could not combine all biomarkers in a common model, separate nomograms demonstrating separate scales did not provide any further contribution to comparing each marker. We expect our results as a preliminary step to guide future prospective studies involving larger, more homogeneous patient cohorts, where validation and reliability analyses can be performed to enable clinical implementation. Therefore, we decided to exclude the nomogram from our final manuscript to align with the study's objectives.

Finally, we would like to highlight that many similar studies in the literature present data using various traditional statistical methods without achieving clinical utility due to a lack of validation or standardization. By focusing on the individual power of these biomarkers and clearly discussing the limitations and possible challenges, our study aims to provide a different point of view for future research.

We hope these clarifications address your concerns, and we are open to further suggestions or additional revisions as needed. Once again, we thank you for your valuable insights and for providing us with the opportunity to discuss our study in greater depth.

Reviewer 4 Report

Comments and Suggestions for Authors

This article reports a retrospective study of  a diverse set of tumors treated with immune checkpoint inhibitors.  For these 226  cancers treated at a single institution, standard prognostic biomarkers were assayed at baseline, and prognostic value of biomarkers in univariate and multivariate contexts assessed.   Overall, the motivation and analyses are clearly presented, and results may be of interest to researchers in the field.  The manuscript is statistically sound with some minor issues/questions. 

MAJOR COMMENTS

The axis labels in Figure 1 are illegible.  They should be made legible.

Line 135:  The IQR is given as 65 years, but in fact this seems to be the range, not the IQR.

The reproducibility of the results presented may depend on the reproducibility of the individual assays that are being studied.  More information about previous studies of the reproducibility of these biomarkers would improve the manuscript.  Also, some discussion of the reproducibility across tumor type/cell type would be helpful.

For section 2.2, please describe any quality control or chart review done on this database to ensure quality.

Line 113, is the “maximally selected rank statistics” the same as the maximum log-rank statistic?  If not, could the difference be described briefly?

The range of 11 years from 2012 to 2023 might raise some concerns about changes in treatment over those years or in patient characteristics creating potential for confounding or bias.  Did the investigators look at controlling for year or time?

As described in section 3.1, the tumor types are quite diverse.  Yet in the models the stages of the tumors are included.  Isn’t the prognosis for these different stages (II/III/IV) quite different in these different tumor types?  If so, there may not be an easy fix, but perhaps this could be listed as a limitation.

Are the “LR’s” in Table 3 under model performance the likelihood ration tests and, if so, are they comparing to the null model or to the model with the selected covariates?  This should be stated clearly somewhere. 

MINOR COMMENTS

It seemed that LDH and CRP were used before defined in the abstract.

Author Response

MAJOR COMMENTS

Comment:1.The axis labels in Figure 1 are illegible.  They should be made legible.

Response 1: Figure 1 Axis Labels:
We appreciate your observation regarding the axis labels in Figure 1. We have increased the resolution and adjusted the font size to ensure that the labels are clear and legible.

Comment: 2: Line 135:  The IQR is given as 65 years, but in fact this seems to be the range, not the IQR.

Response 2: Line 135 - IQR Value:
Thank you for catching this error. The value provided was indeed the range, not the interquartile range (IQR). We have corrected this mistake in the manuscript and appreciate your attention to detail.

Comment 3: The reproducibility of the results presented may depend on the reproducibility of the individual assays that are being studied.  More information about previous studies of the reproducibility of these biomarkers would improve the manuscript.  Also, some discussion of the reproducibility across tumor type/cell type would be helpful.

For section 2.2, please describe any quality control or chart review done on this database to ensure quality.

Response 3: Section 2.2 - Quality Control:
To address your concern regarding data reproducibility, we added details of our data collection and verification process to the manuscript.

Comment 4: Line 113, is the “maximally selected rank statistics” the same as the maximum log-rank statistic?  If not, could the difference be described briefly?

Response 4: Line 113 - Maximally Selected Rank Statistics:
The maximally selected rank statistics were calculated according to the maximum rank statistics with the minimum p-value, as you indicated.

Comment 5: The range of 11 years from 2012 to 2023 might raise some concerns about changes in treatment over those years or in patient characteristics creating potential for confounding or bias.  Did the investigators look at controlling for year or time?

As described in section 3.1, the tumor types are quite diverse.  Yet in the models the stages of the tumors are included.  Isn’t the prognosis for these different stages (II/III/IV) quite different in these different tumor types?  If so, there may not be an easy fix, but perhaps this could be listed as a limitation.

Response 5:  Range of Study Years, Tumor Stage Diversity:
The wide range of years could introduce potential bias. However, it is worth noting that only eight patients in the study were included prior to 2017, and all of these were diagnosed with melanoma. For this group, the available treatment options in our country have not significantly changed to affect clinical outcomes. Regarding patient stages, 95% of the cohort were metastatic at the time of inclusion, and tumor stage was included as a confounder in the multivariate analysis. We acknowledge that this diversity remains a limitation and have addressed this in the discussion.

Comment 6: Are the “LR’s” in Table 3 under model performance the likelihood ration tests and, if so, are they comparing to the null model or to the model with the selected covariates?  This should be stated clearly somewhere. 

Response 6: Table 3 - LR Explanation:
The details on LR values reported in Table 3 are explained. We have updated the table legend and relevant sections of the manuscript to clarify this point.

MINOR COMMENTS

Comment 7: It seemed that LDH and CRP were used before defined in the abstract.

Response 7: Definition of LDH and CRP in Abstract:
We have edited the abstract to ensure that LDH and CRP are properly defined upon their first mention, as suggested.

We are grateful for your careful review and insightful suggestions, which have significantly enhanced the clarity and robustness of our manuscript. We hope that our revisions address your concerns, and we remain open to any additional feedback or recommendations.